# Effectiveness and Safety of SARS-CoV-2 Vaccination in HIV-Infected Patients—Real-World Study

**DOI:** 10.3390/vaccines11050893

**Published:** 2023-04-24

**Authors:** Monika Bociąga-Jasik, Martyna Lara, Aleksandra Raczyńska, Barbara Wizner, Stanisław Polański, Ewa Mlicka-Kowalczyk, Aleksander Garlicki, Marek Sanak

**Affiliations:** 1Department of Infectious and Tropical Diseases, Jagiellonian University Medical College, 30-688 Krakow, Poland; 2Department of Infectious Diseases, University Hospital, 30-688 Krakow, Poland; 3Department of Internal Medicine and Gerontology, Jagiellonian University Medical College, 30-688 Kraków, Poland; 4Division of Molecular Biology and Clinical Genetics, Department of Medicine, Jagiellonian University Medical College, 31-066 Kraków, Poland

**Keywords:** HIV, COVID-19, vaccination, immunogenicity

## Abstract

The development of COVID-19 vaccines has been a triumph of biomedical research. However, there are still challenges, including assessment of their immunogenicity in high-risk populations, including PLWH. In the present study, we enrolled 121 PLWH aged >18 years, that were vaccinated against COVID-19 in the Polish National Vaccination Program. Patients filled in questionnaires regarding the side effects of vaccination. Epidemiological, clinical, and laboratory data were collected. The efficacy of COVID-19 vaccines was evaluated with an ELISA that detects IgG antibodies using a recombinant S1 viral protein antigen. The interferon-gamma release assay (IGRA) was applied to quantitate interferon-gamma (IFN-γ) to assess cellular immunity to SARS-CoV-2. In total, 87 patients (71.9%) received mRNA vaccines (BNT162b2-76 (59.5%), mRNA-1273- 11 (9.1%)). A total of 34 patients (28.09%) were vaccinated with vector-based vaccines (ChAdOx Vaxzevria- 20 (16.52%), Ad26.COV2.S- 14 (11.6%)). A total of 95 (78.5%) of all vaccinated patients developed a protective level of IgG antibodies. Only eight PLWH (6.6%) did not develop cellular immune response. There were six patients (4.95%) that did not develop a cellular and humoral response. Analysis of variance proved that the best humoral and cellular response related to the administration of the mRNA-1273 vaccine. COVID-19 vaccines were found to be immunogenic and safe in PLWH. Vaccination with mRNA vaccines were related to better humoral and cellular responses.

## 1. Introduction

In December 2019, a group of patients with an unknown cause of interstitial pneumonia was reported by local health authorities in Wuhan City, Hubei Province, China [1]. The virus turned out to be highly contagious and quickly spread to many parts of the world, reaching the size of a pandemic as announced by the World Health Organization (WHO) on 11 March 2020 [2]. Up to September 2022, over 614 million cases were confirmed globally and over 6 million people died because of SARS-CoV-2 infection.

HIV infection causes distraction of the immune system, mostly affecting cell-mediated immunity by death of CD4 T-cells and impairment of helper T-cells which could theoretically affect the course of other infectious diseases and vaccinations as well. Current data suggest that, especially in developed countries, there are no differences in the clinical outcomes of COVID-19 among people living with HIV (PLWH) and the general population [3]. However, those with a detectable HIV viral load and comorbid conditions might be at a higher risk of severe COVID-19 [4]. Specific non-infectious comorbidities especially diabetes mellitus, circulatory diseases, and renal failure are undoubtedly much more common among PLWH compared to the general population. Therefore, higher risk of hospitalization and worse disease prognosis may be the result of a combined effect of HIV-related chronic diseases, instead of HIV infection per se. On the other hand, HIV infection might lead to COVID-19-related death that is not associated with such comorbidities [5]. HIV-related factors, such low CD4 T-cell counts, detectable viral load, opportunistic infections, or neoplastic diseases are independent risk factors for severe COVID-19 [6]. Limited data are available on post-acute sequelae of SARS-CoV-2 (PACS) in this group, but some studies indicate that HIV infection strongly predicts the presence of this complication [7].

Immunocompromised patients are less likely to achieve a sufficient immune response to COVID-19 vaccination. In a systematic analysis, the rates of seroconversion after two COVID-19 vaccine shots (pooling across all studies and platforms) were 99% (95% CI 98–100) for people who are immunocompetent, 92% (88–94%) for patients with solid organs cancer, 78% (69–95) for patients with immune-mediated inflammatory diseases, 64% (50–76) for patients with hematological cancer, and 27% (16–42) for patients after transplantation [8]. This fact is worth concern because immunosuppression is connected with persistent SARS-CoV-2 infection and mutations that might escalate virus transmissibility or promote escape from vaccine protection [9].

PLWH are claimed to not respond to some vaccines as effectively as non-HIV-infected individuals. This has been observed in multiple studies in relation to hepatitis B, yellow fever, diphtheria, tetanus, and poliomyelitis vaccines. The diminished vaccine immunogenicity in this group is postulated to by caused by immune dysregulation and persistent inflammation and is more pronounced in patients with lower CD4/CD8 ratios [10].

Owing to unprecedented international effort, several effective and safe COVID-19 vaccines have been developed in less than a year after the pandemic started. Currently, four vaccines are widely used in Poland. Of these, two are mRNA vaccines- BNT162b2 (Comirnaty, manufactured by Pfizer/BioNTech) and mRNA-1273 (Spikevax by Moderna). Their mechanism of action involves introduction of a specific mRNA encapsulated in lipid nanoparticles into the host cell, translation by the ribosomes, and production of SARS-CoV-2 Spike protein which elicits an immune response [11]. The two others are adenovirus vaccines- ChAdOx1 nCoV-19 (Vaxzevria by AstraZeneca) and Ad26.COV.2-S (Jcovden by Janssen). They use replication deficient adenoviruses as vectors engineered to express the SARS-CoV-2 spike protein [12].

PLWH were vastly underrepresented in registration studies of mRNA vaccines. A positive HIV test was an exclusion criterion in all phases of the mRNA-1273 registration study (Spikevax, ClinicalTrials.gov Identifier: NCT04283461, accessed on 25 February 2020, NCT04405076, accessed on 28 May 2020 and NCT04470427, accessed on 14 July 2020) as well as Phases 1 and 2 of the BNT162b2 registration study (ClinicalTrials.gov Identifier: NCT04368728, accessed on 30 April 2020). HIV-infected people were eligible for Phase 2/3 of the latter and constituted 196 out of 44,000 participants enrolled in the study [13].

Safety and reactogenicity of ChAdOx1 nCoV-19 vaccine in HIV-infected individuals was assessed in a sub-study within phases 1B/2 and 2/3 of the AstraZeneca vaccine [14,15]. The Ad26.COV.2-S vaccine trial had the largest HIV representation from the aforementioned vaccines and was able to estimate vaccine efficacy against moderate to severe COVID-19 in a subgroup analysis in PLWH [16].

Several subsequent studies evaluated the effectiveness and safety of COVID-19 vaccines on less numerous groups of HIV-infected participants. Most of them included patients with good HIV control, but mounting evidence generally suggest a less robust response, only partially explained by mere CD4 count at the time of vaccination [14,17,18,19,20].

Real-world studies are still needed to assess the complexity of humoral and cell-mediated responses to vaccines in PLWH, especially in relation to local health policy regarding antiretroviral treatment and COVID-19 vaccination programs. This information is essential for the development of appropriate interventions and to address disparities among this group.

## 2. Material and Methods

The Outpatient Clinic of the University Hospital in Krakow for HIV-infected people serves 1618 ambulatory patients. Enrollment into the study was offered to adult HIV-infected patients (>18 years old) vaccinated against COVID-19 in the Polish National Vaccination Program. Patients with active SARS-CoV-2 infection were excluded from the study. This observational study was based on standard clinical care. All the authors were granted permission to access the medical database as regular employees of the University Hospital. The study was approved by the Jagiellonian University Ethics Committee, decision number 1072.6120.278.2020 and written informed consent was obtained from all the participants.

Patients were asked to fill in a questionnaire regarding local and systemic side effects connected with vaccination. Epidemiological (age, sex, self- defined route of HIV transmission), clinical, and laboratory data including HIV course, CD4-T cell counts, HIV viral load, antiretroviral therapy, and concomitant diseases were taken from the hospital medical electronic database. All data were anonymized.

Serum, EDTA (ethylenediaminetetraacetic acid) plasma, and whole-blood specimens were obtained for measuring SARS-CoV-2-specific antibodies and IGRAs, respectively. Post-vaccine immunization was evaluated with an ELISA that detects IgG antibodies using a recombinant S1 viral protein antigen (Anti-SARS-CoV-2 IgG ELISA, Euroimmun, Luebeck, Germany). The results are expressed as standardized binding antibody units (BAU/mL). A positive humoral response was defined as the level of specific IgG exceeding 30 BAU/mL. Diagnostic values of specific IgG assay were thoroughly evaluated by the Food and Drug Administration, Silver Spring, MD, USA. The assay specificity is 99.6% and sensitivity 94.4%.

The interferon-gamma release assay (IGRA) was applied to quantitate interferon-gamma (IFN-γ) release by SARS-CoV-2-specific T-cells and to assess cellular immunity to SARS-CoV-2. The assay was carried out according to the manufacturer’s instructions (Quan-T-Cell SARS-CoV-2 stimulation IGRA ELISA, Euroimmun, Luebeck, Germany) and expressed in mIU/mL following subtraction of a non-stimulated IFN-γ release in a blank sample. In the IGRA, each study subject simultaneously had three blood samples collected into dedicated tubes. The first was for an unstimulated basal production of IFN-gamma, the second contained a recombinant S1 fragment of SARS-CoV-2 spike protein antigen, and the third had a mitogen stimulating the maximal production of IFN-γ added. The IFN-γ concentrations were interpreted as follows: IFNγ < 100 mIU/mL was considered negative, 100–200 mIU/mL borderline, and >200 mIU/mL as positive. Peripheral blood lymphocytes (B-cell and T-cells were counted using flow cytometry and directly labeled antibodies multicolor mixes (FACSLyric and BD Multitest reagents, BD Biosciences, Franklin Lanes, NJ, USA).

Statistical analysis was performed using R statistical software (R Foundation for statistical computing, Vienna, Austria). Multivariate analysis of variance of IgG or IFN-γ release as dependent variables included type of the vaccine, sex, age, time from vaccination, and lymphocyte subpopulation counts at the time of vaccination as well as CD4-T-cell nadir. Due to specific IgG and IFN-γ levels departed from the normal distribution, analysis of variance was carried out on log-transformed data, which is a valid statistical test for a small size of sample. Pair-wise differences between the vaccine types were estimated using the least significant difference (LSD) post hoc test. Spearman’s rank correlation coefficient was used to assess the impact of CD4 nadir on vaccine response.

## 3. Results

### 3.1. Study Population

In total, 121 people living with HIV (PLWH) aged 40.3 years on average (min. 22–max. 73), including 6 females (4.95%) and 115 (95.04%) males fully vaccinated between January 2021 and August 2021 were investigated. A total of 32 (26.4%) of them had other concomitant diseases, such as hypertension (11.57%), other cardiovascular diseases (3.3%), diabetes mellitus (1.6%), obesity (5.7%), chronic hepatitis (2.4%), liver cirrhosis (1.65%), chronic kidney diseases (2.4%), and alcohol abuse (2.4%). The general characteristics of the study group are presented in Table 1. The mean time from HIV diagnosis to COVID-19 vaccination was 6.4 years (min. 0.5–max. 22). All of the investigated patients were on antiretroviral therapy (ART), but one of them had started treatment one week before vaccination. The mean duration of ART was 5 years (min. 0–max. 27). In total, integrase inhibitor-based (InIs) treatment was used in 80.1% of patients, and 19.8% were on a two-drug regimen (2DR), 8.26% of patients were on proteases inhibitor-based (PIs) regimens, and 11.57% on non-nucleoside reverse transcriptase inhibitors (NNRTIs). A total of 112 patients (92.56%) had an undetectable viral load defined as HIV-RNA < 50 copies/mL on vaccination, the mean viral load at COVID-19 vaccination was 5 567 copies/mL (min. 0–max. 246 000). Nadir CD4 T-cell counts < 50/μL had 8 (6.6%) patients and 24 (19.83%) had nadir CD4 T-cell counts < 200/μL. A total of 19 patients (15.7%) had full blown AIDS on diagnosis, and 8 (6.6%) had neoplastic disease in their history. The mean CD4 T-cell counts at COVID-19 vaccination was 878 cells/μL (min. 213–max. 1824). No one in the investigated groups had CD4 T-cell counts < 200/μL, and only 19 (15.7%) patients CD4 T-cell counts ranging from 200 to 500/μL at the time of vaccination. Detailed HIV-related variables are presented in Table 2.

A total of 87 patients (71.9%) received mRNA vaccines, BNT162b2 mRNA vaccine (two shots) was given to 76 patients (59.5%), mRNA-1273 (two shots) to 11 patients (9.1%). A total of 34 patients (28.09%) were vaccinated with vector-based vaccines; 20 patients (16.52%) with ChAdOx Vaxzevria (two shots) and 14 patients (11.6%) with Ad26.COV2.S (one shot). A total of 21 patients (17.3%) reported previous SARS-CoV-2 infection, min. 50–max. 515 days before vaccination.

### 3.2. Humoral Immune Response to SARS-CoV-2 Vaccination

A total of 95 (78.5%) of all vaccinated patients developed a protective level of anti-S IgG antibodies.

However, anti-S IgG antibodies were not detected in 13.7% (n- 12/87) of PLWH vaccinated with mRNA vaccines and 41.17% (n- 14/34) vaccinated with vector-based vaccine. A total of 12 patients (15.75%) of the 76 vaccinated with the BNT162b2 vaccine did not respond. All patients vaccinated with mRNA-1273 had an adequate anti-S IgG antibodies level. A total of 40% (n- 8/20) of recipients of ChAdOx Vaxzevria vaccine and 42.8% (n- 6/14) Ad26.COV2.S of had anti-SARS-CoV-2-S 1 IgG < 30 BAU/mL.

### 3.3. Cellular Immune Response to SARS-CoV-2 Vaccination

Of all 121 participants, only eight HIV-infected patients (6.6%) did not develop cellular immune response, 4.59% (n- 4/84) vaccinated with mRNA vaccines and 11.76% (n- 4/34) with vector-based vaccines. This was the same for 4 of 76 patients (5.2%) vaccinated with BNT162b2, 2 out of 20 patients (10%) vaccinated with ChAdOx Vaxzevria, and 2 out of 14 patients (14%) who received Ad26.COV2.S. All patients vaccinated with mRNA-1273 vaccine developed cellular immune response.

Of the patients, six (4.95%) did not develop a cellular and humoral response, 3.44% (n- 3/87) in the m-RNA vaccine group, BNT162b2—4.1% (n-3/76 pts), mRNA-1273—0% (0/11); and 8.82% (n- 3/34) in the vector-based vaccine group (ChAdOx Vaxzevria- 10%, (n- 2/20 pts), Ad26.COV2.S– 7.1% (n- 1/14)).

Non-responder individual case consideration revealed that all of them were older than 60 years old, with at least one severe concomitant disease, and with low CD4 T-cell nadir (min. 10–max. 60 cells/µL, mean. 45 cells/µL). One of the non-responders was 63 y.o., MSM, diagnosed with HIV on May 2013, CD4 T-cell nadir—22 cells/µL (4%), CD4/CD8—0.04, HIV-RNA on diagnosis—88,000 copies/mL and PCP at the time of diagnosis. His concomitant diseases were chronic hepatitis type B with liver cirrhosis, diabetes mellitus, osteoporosis, and hypertension, with a CD4 T-cell on vaccination—224 cells/µL. CD4/CD8—0.23, HIV-RNA—undetectable. The vaccination type in his case was the vector-based vaccine ChAdOx Vaxzevria.

### 3.4. Assessment of Immunization following Vaccination

The immunogenicity of COVID-19 vaccines in PLWH is summarized in Table 3.

The immune response to vaccination with COVID-19 vaccines in the investigated group of HIV-infected patients was analyzed using a multivariate analysis of variance using either specific anti-S IgG levels or IGRA test as dependent variables. Co-variables related to the immunological status are presented in Table 4.

These variables were included in the multivariate analysis and did not contribute to the differences caused by a type of the vaccine used. *p*-values were from 0.08 for lymphocytes B (CD19+) to *p* = 0.14 for CD4 T-cell in the case of IgG. No cellular fraction of peripheral blood lymphocytes correlated with IFN-γ (*p* > |0.2|)

Multivariate analysis of variance confirmed that the levels of IgG and IFN-γ release after vaccination were significantly influenced by the type of vaccine administered regardless of the previous COVID-19 history (i.e., whether the patient had confirmed COVID-19 infection in the past). Analysis of variance proved that the best humoral and cellular response related to administration of the mRNA-1273 vaccine. The CD4 T-cell count at the time of vaccination as well as the CD4 T-cell nadir had no impact on the specific IgG or IFN-γ levels. There was no effect of the patients’ age and sex on post-vaccination immunity parameters. However, as expected within a period of an average of 115 ± 44 days, specific IgG levels significantly decreased (r = −0.489), as well as the capacity to produce IFN-γ by sensitized lymphocytes (r = −0.245). Table 5 shows averages with standard deviations within the patient’s groups stratified by the vaccine used or by a previous COVID-19 infection. The first four rows document a highly significant contribution of the type of vaccine on IgG and IFN-γ. The next three rows document the absence of a detectable effect of a previous infection with the virus (COVID-19 infection preceding vaccination). The last row of this table shows a lack of interaction between these two terms, i.e., no impact of the type of vaccine on specific IgG or IFN-γ regardless of previous COVID-19 infection.

The results are shown in Table 5, and Figure 1 and Figure 2.

No correlation was found between the CD4 T-cell nadir and immunological response to vaccination in terms of anti-S IgG (r = −0.146, *p* = 0.113) and IFN-γ levels (r = −0.027, *p* = 0.767).

### 3.5. Safety of Vaccination in the Investigated Cohort

A total of 85 patients (70.24%) reported adverse events. The most common were local reaction 81/121 (67%), headache- 38/121 (31.4%), and fever 24/121 (19.83%). No serious adverse events were reported in all the vaccinated participants. In patients who had two shots, systemic adverse events were more common following the first shot.

## 4. Discussion

The current evidence suggests that people with HIV infection, especially those on stable ART therapy are not at higher risk of SARS-CoV-2 infection or severe course of the disease as the general population. However, in the setting of viremia, immunosuppression, and other chronic diseases, PLWH might have an increased risk of COVID-19-related clinical complications and should be monitored closely [4,5,6]. We present a real-world study reporting on the efficiency and immunogenicity of vaccines against SARS-CoV-2 among PLWH. We analyzed the data approximately 121 patients with HIV that were vaccinated in the Polish National Vaccination Program to evaluate the response to mRNA and vector-based vaccines. The analyzed group was relatively young and most of the patients were on effective, stable HIV therapy with an undetectable viral load and high CD4 cells level, which reflects the population of diagnosed PLWH on active antiretroviral treatment in high income countries. However, some of the patients were diagnosed in advanced stages of HIV, with low CD4 T-cell counts at diagnosis, and a history of AIDS-defining illnesses, including neoplastic diseases.

There are no doubts that PLWH should receive COVID-19 vaccines, although the efficacy data are conflicting. Our results confirm that the vaccines currently used in COVID-19 prevention can evoke a strong humoral- and cell-mediated response in PLWH. There was no correlation in the present study between the level of CD4 T-cells at the moment of vaccination as well as CD4 T-cell nadir and the specific IgG or IFN-γ levels, but it must be stressed that no one in the investigated group had CD4 T-cell counts < 200/μL. Individual case considerations indicate that patients older than 60 years, with severe concomitant diseases, and low CD4 T-cell nadir counts who are at high risk of fatal course of COVID-19 can fail vaccination. It needs further studies which can guide the plans for boosters, and select the group of PLWH who require measurement of IgG antibodies and IFN-γ.

More than 78% of our HIV-positive patients developed protective levels of IgG antibodies. Overall, higher rates were reported by Chun et al. in a large meta-analysis, where only 3 of the 28 studies noted seroconversion rates below 85% [21]. Higher rates (95.2%) were also reported in the Italian cohort of HIV-infected patients [22].

However, antibody response after vaccination also varies among HIV-negative individuals depending on many factors, including age, sex, ethnicity, and comorbid conditions [23]. Our cohort was relatively homogeneous and young, and comprised mostly men, most of whom had no concomitant diseases. Analyzing the obtained results, it should be taken into consideration that in the investigated group, all patients had CD4 counts >200 cells/μL, and as many as 84.3% had CD4 counts >500 cells/μL. Some other researchers proved that PLWH with CD4 counts <200 cells/μL. had significantly lower humoral and cellular response [24]. A total of 32 patients (26.4%) had other chronic diseases and they were less likely to achieve a sufficient response to COVID-19 vaccination.

In our study group, 71% of the patients were vaccinated with an mRNA vaccine and the best response, in terms of IgG titers and interferon-gamma release, was associated with the mRNA-1273 vaccine. These findings are partly in line with the results of a longitudinal analysis conducted by Zhang and colleagues, where mRNA vaccines, especially mRNA-1273, induced a more robust humoral and cell-mediated response than the other vaccines [25].

A similar conclusion is presented in the medRxiv preprint, where having received two doses of vector-based ChAdOx1 vaccine was associated with lower humoral responses than heterologous or autologous mRNA vaccine regimens [26]. To the best of our knowledge, the superiority of the mRNA-1273 vaccine in PLWH has not been unequivocally proven clinically.

A total of 70% of our patients reported side effects following vaccination. Consistent with the literature, the vast majority of them were mild, self-limited, and restricted to the local reaction [27].

The most important limitation of our study is the lack of a control group. However, the large trial assessing the immunogenicity of mRNA-1273, BNT162b2, and Ad26.COV2.S vaccines in healthy individuals yielded results analogous to ours, namely the highest antibody concentrations and T-cell response after the mRNA-1273 vaccine and generally higher effectiveness of mRNA vaccines [28]. As a result of the study design, we were also not able to evaluate the efficiency of vaccination in the group of patients with uncontrolled HIV infection and AIDS-defining conditions. Some data show that vaccination may not be as effective in PLWH with very low CD4 cells count or detectable HIV viral load, and monitoring is especially important in these cases [29]. In the investigated group, the past COVID-19 infection was determined by the questionnaire, not based on the anti-NP antibodies.

## 5. Conclusions

COVID-19 vaccines were found to be immunogenic and safe in PLWH.Vaccination with mRNA vaccines (BNT162b2; mRNA-1273) related to better humoral and cellular responses.For HIV patients it is important to collect data on immunogenicity pointing to a higher effectiveness of mRNA vaccines.The CD4 T-cell count as well as CD4 T-cell nadir had no impact on the specific IgG or IFN-γ levels, however, based on the selected case analysis, we suggest that PLWH with low CD4 T-cell nadir, low CD4 T-cell count (~200 cell/μL), and comorbidities would benefit from monitoring vaccine responsiveness.

## Figures and Tables

**Figure 1 vaccines-11-00893-f001:**
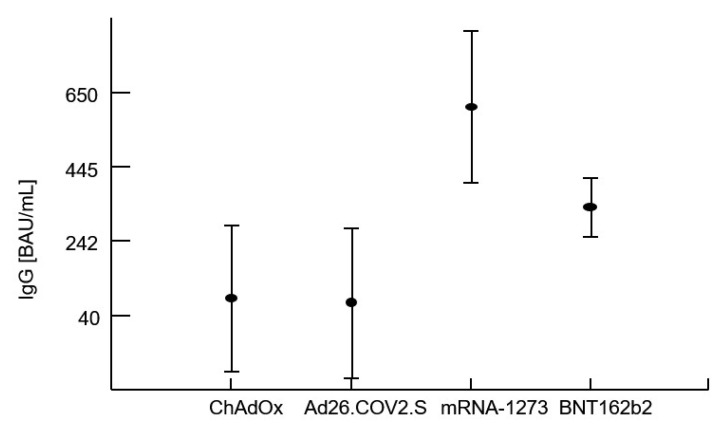
Comparison of specific anti-S IgG following vaccination with four different vaccines. The average binding antibody units (BAU) are plotted using post hoc means adjusted by covariates of the multivariate analysis procedure and have standard error of the mean whiskers.

**Figure 2 vaccines-11-00893-f002:**
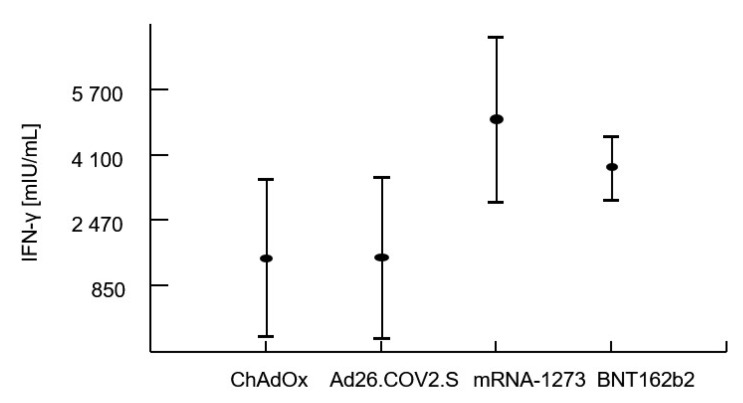
Comparison of IFN-γ levels in an interferon-gamma release assay (IGRA) following vaccination with four different vaccines. The average IFN-γ concentrations are plotted using post hoc means adjusted by covariates of the multivariate analysis procedure and have standard error of the mean whiskers.

**Table 1 vaccines-11-00893-t001:** Characteristics of the patients.

Variable	
Male/Female (n)	115/6
Age (years)	
mean ± SD	40.3 ± 11.0
range (min–max)	22–73
Co-morbidities (n, %)	32 (26.4)
Hypertension	14 (11.57)
Cardiovascular diseases	4 (3.3)
Diabetes mellitus	2 (1.6)
Obesity (BMI > 30)	7 (5.7)
Chronic hepatitis	3 (2.4)
Liver cirrhosis	2 (1.65)
Chronic kidney diseases	3 (2.4)
Alcohol abuse	3 (2.4)
Days after vaccination	
mean ± SD	115.1 ± 44.2
range (min–max)	60–259
Previous SARS-CoV-2 infection (n, %), days before vaccination	21 (17.3%); min. 50–max. 515 days (mean—285 days)
Type of vaccine (n):	
BNT162b2	76
mRNA-1273	11
Ad26.COV2.S	14
ChAdOx nCoV-19	20

**Table 2 vaccines-11-00893-t002:** HIV-related variables, *n* = 121.

Time from Diagnosis (Years)	
mean ± SD	6.4 ± 5.4
median [Q1–Q3]	4.8 [2.6–9.0]
range [min–max]	0.2–27.7
Route of transmission	
MSM, n (%)	101 (83.47)
HET, n (%)	6 (4.95)
IDU, n (%)	4 (3.3)
AIDS on diagnosis, n, (%)	19 (15.7)
Neoplastic diseases in the history, n (%)	8 (6.6)
Nadir CD4 T-cell counts /μL	min. 7–max. 1288 (mean—435)
<50, n (%)	8 (6.61)
50–200, n (%)	24 (19.83)
>200, n (%)	88 (72.73)
Viral load (HIV-RNA- copies/mL) on diagnosis	min. 7000–max. 10,000,000 (mean—369,485)
CD4 T-cell counts/μL on diagnosis	min. 7–max. 1711 (mean—492)
CD4 T-cell counts/μL on COVID-19 vaccination	min. 213–max. 1824 (mean—878)
<200, n (%)	
200–500, n (%)	0
>500, n (%)	19 (15.70)
	102 (84.30)
Viral load (HIV-RNA- copies/mL) on COVID-19 vaccination	min.0–max. 246,000 (mean—5567)
Number of patients with undetectable viral load defined as HIV-RNA < 50 copies/mL, n (%)	
	112 (92.56%)
Integrase inhibitor-based treatment	97 (80.1)
2DR	24 (19.8)
Protease inhibitor-based treatment	10 (8.26)
NNRTIs–based	14 (11.57)

**Table 3 vaccines-11-00893-t003:** Immunogenicity of COVID-19 vaccines in PLWH (*n* = 121).

Variables	
IgG- (BAU/mL)	
median [Q1–Q3]	95.7 [37.9–324.6]
range (min–max)	1.6–1400.0
IFN gamma (mIU/mL)	
median [Q1–Q3]	1927.7 [913.1–4247.3]
range (min–max)	45.0–9069.4

**Table 4 vaccines-11-00893-t004:** Co-variables related to the immunological status (*n* = 121).

Variables	
Limf T_CD3	
median [Q1–Q3]	1489 [1196–1853]
range	497–3171
NK	
median [Q1–Q3]	263 [181–362]
range	66–801
Limf B_CD19	
median [Q1–Q3]	188 [128–245]
range	30–550
LT_CD4	
median [Q1–Q3]	687 [521–893]
range	192–1768
LT_CD8	
median [Q1–Q3]	604 [470–857]
range	180–2453

**Table 5 vaccines-11-00893-t005:** Multivariate analysis of variance; the impact of the type of vaccination and the history of COVID-19 on the humoral and cellular response. Anti-S IgG levels in BAU/mL, IFN-γ in mIU/mL.

	Anti-S IgG Mean, SD	*p*-Value	IFN-γ Mean, SD	*p*-Value
Type of vaccine		0.000		0.014
BNT162b2	298.58 (357.18)		3261.35 (2757.56)	
MRNA-1273	508.56 (368.83)		5666.41 (3217.80)	
Ad26.COV2.S	29.92 (34.07)		1326.99 (1855.14)	
ChAdOx nCoV-19	85.85 (124.50)		1433.46 (1376.60)	
History of COVID-19		0.245		0.816
YES	417.05 (344.34)		3707.84 (2621.33)	
NO	210.85 (316.36)		2897.42 (2859.60)	
Type of vaccine*COVID-19		0.249		0.597

## Data Availability

Data supporting reported results can requested to the authors.

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
