# Peer review of "Effectiveness and Safety of SARS-CoV-2 Vaccination in HIV-Infected Patients—Real-World Study"

_vaccines, 2023, doi:10.3390/vaccines11050893_

Round 1

Reviewer 1 Report

In the article entitled "Effectiveness and safety of SARS-CoV-2 vaccination in HIV infected patients- real-world study" by BociÄ…ga-Jasik M et al; analyzed the effectiveness of SARS-CoV-2 vaccines in HIV patients and found that mRNA vaccines work better and generated both cellular and humoral responses in higher degree compared to non mRNA vaccines. Study also finds that elderly patients with comorbidities did not respond well to both vaccines. The mRNA vaccines were also shown higher IgG and IFN-Y compared to vector-based vaccines. This manuscript will be able to fill the gaps in the understanding of vaccine responses in HIV patients and will help to develop vaccines for these minority groups in future. 

Reviewer 2 Report

The authors have studied the effectiveness and saftety of SARS -CoV-2 vaccines on patients infected with HIV. Though the study is informative and shows a sign of relief among HIV infected inviduals, I have the following suggestions/comments:

1. As the authors have mentioned already, control group is missing. The lack of control group overall makes the study redundant.

2. The authors have concluded mRNA-1273 vaccine appears to be more effective. However the sample size is too small to reach to any conclusion from this study. Though the authors claim that their observation correlates with other studies, I would still suggest that the sample size needs to be increased before reaching to any such conclusion.

3. The interferon-gamma release assay can be affected by HIV infection. I am not very convinced if this assay can be used to assess cell mediated immunity in HIV infected patients. The ideal condition would be to compare the interferon gamma release before and after vaccination.

4. If the duration since SARS-CoV-2 vaccination was taken into the account for each vaccine category? If the booster doses were taken into account?

Round 2

Reviewer 2 Report

The authors have either made the suggested changes or have explained the short comings of the study that makes the manuscript easy to read and convincing.